# Fatty-Acid-Based Membrane Lipidome Profile of Peanut Allergy Patients: An Exploratory Study of a Lifelong Health Condition

**DOI:** 10.3390/ijms24010120

**Published:** 2022-12-21

**Authors:** Elisabetta Del Duca, Anna Sansone, Mayla Sgrulletti, Federica Di Nolfo, Loredana Chini, Carla Ferreri, Viviana Moschese

**Affiliations:** 1Pediatric Immunopathology and Allergology Unit, Policlinico Tor Vergata, University of Rome Tor Vergata, Viale Oxford 81, 00133 Rome, Italy; 2ISOF, Consiglio Nazionale delle Ricerche, Via Piero Gobetti 101, 40129 Bologna, Italy; 3PhD Program in Immunology, Molecular Medicine and Applied Biotechnology, University of Rome Tor Vergata, Via Montpellier 1, 00133 Rome, Italy; 4Lipidomic Laboratory, Lipinutragen Srl, Via di Corticella 181/4, 40128 Bologna, Italy

**Keywords:** peanut allergy, allergic inflammation, red blood cell, membrane lipidome, ω-6 fatty acids, ω-3 fatty acids, polyunsaturated fatty acids

## Abstract

Peanut allergy is a lifelong, increasingly prevalent, and potentially life-threatening disease burdening families and communities. Dietary, particularly polyunsaturated fatty acids (PUFAs), intakes can exert positive effects on immune and inflammatory responses, and the red blood cell (RBC) membrane lipidome contains stabilized metabolic and nutritional information connected with such responses. The fatty-acid-based membrane lipidome profile has been exploratorily evaluated in a small cohort of patients (eight males and one female, age range 4.1–21.7 years old, body mass index BMI < 25) with angioedema and/or anaphylaxis after peanut ingestion. This analysis was performed according to an ISO 17025 certified robotic protocol, isolating mature RBCs, extracting membrane lipids, and transforming them to fatty acid methyl esters for gas chromatography recognition and quantification. Comparison with a group of age- and BMI-matched healthy individuals and with benchmark interval values of a healthy population evidenced significant differences, such as higher levels of ω-6 (arachidonic acid), lower values of ω-3 eicosapentaenoic acid (EPA) and docosahexaenoic acid (DHA), together with an increased ω-6/ω-3 ratio in allergic patients. A significant inverse correlation was also found between specific immunoglobulin E (IgE) levels and ω-6 di-homo-gamma-linolenic acid (DGLA) and total PUFAs. Results of this preliminary study encourage screenings in larger cohorts, also in view of precision nutrition and nutraceuticals strategies, and stimulate interest to expand basic and applied research for unveiling molecular mechanisms that are still missing and individuating treatments in chronic allergic disorders.

## 1. Introduction

Peanut allergy (PA) is one of the most common food allergies in the pediatric age, affecting approximately 1–3% of children in Western countries, and the prevalence has been increasing in the last decades worldwide [1,2]. Compared with other food allergies, PA is usually lifelong and associated with higher rates of accidental exposure, severe reactions, and potentially fatal anaphylaxis. This, in turn, has a significant impact on the management of the patient, also from a nutritional point of view. The immune system and impaired immune maturation are involved in tolerance breakdown and the development of immune-mediated diseases such as food allergies [3]. The involvement of lipids is progressively recognized both as basic constituents of cellular structures involved in immune regulation and components of diets [4].

### 1.1. Fatty Acids as Structural and Functional Constituents of Cell Membranes and Relationship with Diet

Fatty acids are distinguished by their structures into saturated, monounsaturated, and polyunsaturated families (SFA, MUFA, and PUFA), with an important dietary dependence on PUFAs, which are not directly produced by human metabolism. The omega-6 (ω-6) and omega-3 (ω-3) precursors, linoleic and alpha-linolenic acids, are processed after intake by desaturase and elongase enzymes to form long-chain PUFAs (LC-PUFAs), as shown in Figure 1 [5].

LC-PUFAs incorporated into membrane phospholipids are crucial for their subsequent enzymatic release and, consequently, the formation of bioactive lipid mediators (prostaglandins, leukotrienes, etc.), playing important and well-known biological roles in the development and regulation of the immune system and inflammatory process [6,7]. It is worth recalling that allergic patients start to exclude food(s) from their diets, and this is an initial step for introducing fewer essential fatty acids (EFAs) or breaking the balance between ω-6 and ω-3 intakes, i.e., between pro- and anti-inflammatory components. Indeed, an optimal ω-6/ω-3 balance realizes the ordered sequence of ω-6 LCPUFAs derived mediators, which initiate the “reactivity” response, followed by mediators derived from ω-3 LCPUFAs called “specialized proresolving mediators” (SPMs), which bring resolution and tissue repair [8,9]. Only with the balanced production of these mediators can the overall cell response occur in a physiological manner. Dietary unbalances are recognized to be risk factors for atopy, asthma, and allergy [2,10], and fatty acid supplementations are proposed for the treatment of allergic conditions [11,12]. On the other hand, the association between ω-3 and ω-6 intakes and clinical improvements in food allergy patients [13] are under discussion and, more importantly, are not combined yet with the examination of the patient’s fatty acids status, *per se* or during supplementation, in particular utilizing erythrocyte membrane lipidome information, which is a reliable mirror of fatty acid intake and metabolism in the body tissues [14].

### 1.2. Fatty-Acid-Based Membrane Lipidome Analysis

Fatty acid detection can be performed in plasma lipids or blood cell lipids, in particular red blood cells (RBCs), providing different types of information. Diet composition of the days before blood withdrawal strongly affects fatty acid plasma levels, in contrast to RBC membranes, where fatty-acid-based phospholipid composition reflects a balance between nutritional and metabolic factors (i.e., fatty acids transformation into phospholipids). Easiness of the sampling and work-up procedures leads to be more prone to choose plasma or whole blood specimens, but knowledge of the importance of membranes for cell signaling and precision medicine applications strongly indicates RBC membrane isolation and analysis to obtain information on the stabilized metabolic and nutritional status of patients [14]. Indeed, the fatty-acid-based membrane lipidome profile of mature RBCs (mean lifetime of 120 days) was developed and used as a nutritional, homeostatic, and metabolic biomarker in several human physiopathological states [15,16]. Certified laboratory protocol with fatty acid identification and quantitation following ISO 17025 international requirements ensures the reliability and repeatability of the results, as well as the quality of the data. With such a procedure, observational clinical studies were performed identifying membrane profiles in different health conditions [17,18,19,20,21,22]. This information opens the way to personalize the approach of membrane lipid therapy, a natural medicine tool effective in several health conditions [23,24].

## 2. Results

With the aim of exploring the relevance of the membrane lipidome profile in food allergy, a cohort of peanut-allergic patients was selected among subjects followed along years at the Pediatric Immunopathology and Allergology Unit of the university hospital. As shown in the diagram of Figure 2, we recruited *n* = 29 subjects with a clear history of angioedema and/or anaphylaxis, and exclusion criteria were applied (body mass index (BMI) > 25, other food allergies, autoimmune diseases, and allergic reactions in the last 18 months). The final cohort presented *n* = 9 patients (age 12 ± 5.6 years old). At the same time, 15 healthy individuals (8 females and 7 males) with BMI < 25 and age = 17 ± 4 years old were selected from the anonymous database available at the Lipidomic Laboratory, for which informed consent for research use was gathered at the moment of blood withdrawal, and respect of EU general data protection regulation 2016/679 (GDPR) guaranteed by the ISO 17025 certification. The controls did not have allergic problems. Mature RBC membrane fatty acid values compared with those of the patients were obtained by the same procedure. The observational study focused on the fatty acid composition of RBC membranes addressing the following key points: (a) the status of ω-6 vs. ω-3 fatty acid residues of the membrane phospholipids as the expression of proinflammatory tendency in allergic patients, and (b) the correlation of membrane lipidome profile features with specific immunoglobulin E (IgE) levels.

Clinical and laboratory data from peanut-allergic patients are described in Table 1. All patients had a history of angioedema and/or anaphylaxis after ingestion of peanuts; however, the recruited patients did not have any allergic reactions in the last 18 months.

In Table 2, the results of the mature RBC fatty-acid-based membrane lipidome analysis are shown; as in previous work, the main fatty acids of the RBC membrane phospholipids are grouped to give a cohort of 10 SFA, MUFA, and PUFA components, the latter ones representing the ω-6 and ω-3 metabolic cascades shown in Figure 1 [15]. Each unsaturated fatty acid of the cohort is recognized by appropriate standards, ensuring that it is not superimposed to geometrical or positional isomers. Calibration with standard references allows to calculate the quantity of each fatty acid in the sample and express it as a relative quantitative amount (% rel. quant.) over the total of the 10 fatty acid quantities. From these values, the ω-6/ω-3 ratio, related to pro- and anti-inflammatory balance [25], and peroxidation index (PI), as a measure of the PUFA contribution to elevating membrane peroxidizability [26,27], were calculated. Each of the 10 fatty acids and related indexes were compared with the 15 age- and BMI-matched healthy controls and with the benchmark of interval values referred to a healthy population described in previous works [15,16,17]. In Table 2, significant changes observed between patients and controls are reported with their *p*-values.

As shown in Table 2, in allergic patients, we found a marked unbalance mainly concerning the PUFA residues of cell membrane phospholipids. Compared with the cohort of healthy controls, the patients showed a significant increase in ω-6 20:4 (arachidonic acid, ARA), PUFA ω-6 and the ratio ω-6/ω-3, whereas ω-3 20:5 (eicosapentaenoic acid, EPA), 22:6 (docosahexaenoic acid, DHA), total PUFA ω-3, and the PI value were significantly diminished. In addition, total saturated fatty acids (SFAs) and stearic acid (18:0) were significantly lower than controls. The distribution of PUFA values and ratio as scattered dot plots showing each patient is presented in Figure 3 in comparison with those found in healthy controls, reporting also the benchmark of interval values in the healthy population. Although the cohort of patients is small, in Figure 3, it is possible to appreciate that their membrane fatty acid values are all positioned outside (lower or higher) both healthy controls and benchmark interval values. It is worth underlining that levels of trans fatty acids isomers of oleic and arachidonic acids were also evaluated since these compounds are available as standard references [17,18] and found to be significantly increased, although within the values reported for the benchmark (≤0.4).

Patients’ clinical and laboratory data (Table 1) were then correlated to membrane FA values (Table 2), resulting in the heatmap shown in Figure 4.

As detailed in Table 3, significant inverse correlations were found for peanut-specific IgE with ω-6 20:3 di-homo-gamma-linolenic acid DGLA and total PUFA (*p*-value = 0.033 and *p*-value = 0.038, respectively).

## 3. Discussion

This is the first study to explore the relevance of the membrane lipidome profile in patients with peanut allergy. We observed a clear unbalance between ω-6 and ω-3 fatty acids in this cohort of patients, mainly due to an increase in ω-6 20:4 (ARA) and a decrease in ω-3 (EPA and DHA). Considering the role of ARA as a precursor of proinflammatory eicosanoids [9], but also as a result of the metabolism of omega-6 linoleic acid taken from the diet (see Figure 1), our data support the synergy of metabolism and diet to build up a specific profile of the cell membranes of peanut-allergic patients [28]. The role of omega-3, both as essential components of the diet and as modulators of the immune and anti-inflammatory processes, is well known [28], and in our cohort, a significant reduction in ω-3 fatty acids involves EPA and DHA. Examining in detail the patients’ values in Figure 3, in nine patients, the DHA values resulted to be in the lower ranges compared with both healthy controls and the benchmark of the healthy population. As shown in Figure 1, EPA and DHA are the members of the ω-3 cascade containing the highest number of double bonds among all PUFAs (see Figure 1). Besides their transformation into bioactive lipid mediators with immune-stimulating and anti-inflammatory activities [12], DHA exerts a greater influence on membrane biophysical properties such as flexibility, fluidity, and thickness [29,30]. It is also clear that the quantity and quality of fat intake directly from seafood (fish, algae) can influence the levels of these fatty acids in cells; however, all patients apparently had regular fish consumption in their nut-free diet, not different from the healthy controls. Another cause of EPA and DHA diminutions can be the efficiency of the metabolic transformations in these patients, as shown in Figure 1. Indeed, humans have a limited capability to synthesize LC PUFA from the precursor alpha-linolenic acid [31], so that daily intakes of 250 mg EPA and DHA are indicated by the most relevant food safety agencies, such as EFSA (European Food Safety Authority) [32]. These preliminary data on ω-3 deficiency must be deepened in a study with larger cohorts.

We also consider it relevant that the ω-6 components came into the scenario of allergic patients, in particular with an increase in arachidonic acid, with known roles in the propagation of inflammatory responses and cellular reactivity. Moreover, ω-6 DGLA, which is a precursor of arachidonic acid and prostaglandins (PG series 1) [33], showed an inverse correlation with specific IgE levels. Literature data connect DGLA supplementation with an increase in PGD1 and improvement in atopic dermatitis, although in an animal model [34]. Moreover, prostaglandin E1 (PGE1) originates from DGLA, and a synthetic analog misoprostol has been reported to modulate histamine release from basophils [35]. It must be underlined that clinical trials on gamma-linolenic acid (GLA) supplementation as a precursor of DGLA (see Figure 1) gave very heterogeneous results, but no data are yet reported on the follow-up of the GLA-DGLA metabolic transformation and incorporation of DGLA in membrane phospholipids. Evidently, the possibility of monitoring DGLA levels in the RBC membrane offers the best strategy to follow up subjects during treatments. Moreover, total PUFA levels were negatively correlated with specific IgE, and both correlations underline the importance of deepening the PUFA status in patients in view of its importance for cell signaling. On the other hand, despite multiple evidence for PUFA relevance in different health conditions [6,7,9,13,33], the punctual follow-up of patients is still missing in clinical approaches.

The exclusion of subjects with a BMI > 25 from our cohort, as well as from the healthy controls, takes into account that overweight and obesity are known to increase arachidonic acid levels in RBC membranes [22,36], and we wanted to exclude interferences from known proinflammatory conditions. In addition, the distance of 18 months from the last allergic episode in patients and the exclusion of allergic conditions in the healthy cohort were carefully checked to eliminate interferences from immune reactivity. It is interesting to observe that in obesity, the membrane fatty acid asset had some common features with that of allergic patients (such as low levels of DHA and increased levels of ARA and omega-6/omega-3 ratio) [22,36], whereas an increase in the SFA/MUFA ratio was not found, which is known to influence membrane properties through an increased rigidity of RBC membranes [37].

The overall oxidative reactivity estimated with the peroxidizability of RBC membranes (peroxidation index, PI; cfr., Table 2) in allergic patients was lower than in controls. However, it must be underlined that we did not perform direct measurements of peroxidation processes through a measure of oxidation metabolites, and this is a limitation of the present study. In our analysis, we also measured trans fatty acid (TFA) isomers of oleic and arachidonic acids, which are known markers of endogenous cis-trans isomerization of the corresponding cis MUFA and PUFA caused by increased free radical production [38,39]. Previously, we reported TFA to increase in the blood cell membranes of children affected by atopic eczema/dermatitis syndrome [40]. In our allergic cohort, we found TFA significantly increased compared with healthy controls (0.35 ± 0.07 vs. 0.18 ± 0.05, *p-*value ≤ 0.0001) [15]; however, the threshold value of the benchmark (0.4%) was not overcome. 

The detection of fatty acids and isomers is one of the various aspects of the analytical protocol that starts from the choice of mature RBC membranes for sampling, as detailed in the experimental part and discussed in previous research papers and reviews [16,20,21,22]. Precision medicine must rely on analytical data not only obtained by protocols unified among laboratories but also certified for the results and reliability by a competent auditing process, usually provided by national bodies of accreditation through compliance with the ISO 17025 regulation. The use of gas chromatographic methodology, with cis and trans fatty acid references and calibration procedures, allows to examine fatty acid levels with the highest molecular identification and sensitive quantification, and this is needed when membrane lipidome profiles are developed to identify disease onset [14,16,18,20]. Moreover, the inclusion of a robotic platform in the certified protocol reduces errors due to manual operations. Our small cohort certainly took advantage of such precise measures to evidence significant results, but we are aware that the sample size must be increased to achieve greater statistical power. Here, we preliminarily showed results of membrane-based diagnostics in peanut allergy patients, supporting the importance of this tool that has reached maturity and technological advancement to serve larger screenings.

Further studies will be able to address an integrated vision in allergic patients in which specific alterations found in RBC membrane lipidome profiles mirror crucial changes in molecular components and related signaling to develop allergic reactivity [41]. Some of these changes are indicated in Figure 5, starting from the formation of the fatty acid pool, influenced by nutritional and metabolic contributions specific to each individual, and the consequences on the composition of membrane phospholipids. Once membranes are formed, they present different qualities and quantities of PUFAs that, in turn, can bring a balance/unbalance of bioactive lipids created in cells after the release of PUFA residues from phospholipids. An increase in ARA and a diminution in DGLA, EPA, and DHA in allergic patients can contribute to an unbalance of signaling with an influence on inflammatory and anti-inflammatory cytokine productions, as well as on a diminution in the protection from antigen-induced activation given by ω-3 proresolving mediators (epoxins, resolvins, protectins) [41], resulting in a general cellular response toward hypersensitivity and augmented IgE production [42]. 

Alteration of the PUFA composition in membranes is basic information to acquire and translate into precision medicine and nutrition strategies. In fact, the type and dosage of fatty acids able to rebalance the molecular status of patients can be personalized for each distinct profile and regularly monitored. This is particularly important in children, in which PUFAs are used to sustain the exponential growth of cells for all tissues, and their exact levels must be determined [43] in order to promptly individuate deficiency or excess with impact on normal tissue functioning [44,45]. Our exploratory study highlights cell membranes for effecting PUFA detection exactly in the active site of their immunomodulatory effects, evidencing molecular mechanisms that are still missing in the evaluation of PUFA treatments in chronic allergic disorders.

## 4. Materials and Methods

### 4.1. Study Subjects

This study was conducted in accordance with the ethical principles of the Declaration of Helsinki and obtained ethical clearance from the ethics committee at Policlinico Tor Vergata, University of Rome Tor Vergata (n.82.21). Informed consent was obtained from each patient or patient’s legal guardians and each of the healthy controls. As reported in Figure 2, a total of 29 Italian native patients with peanut allergy (18 M and 11 F, age range 4.1–21.7 years) were identified at the Pediatric Immunopathology and Allergology Unit, Policlinico Tor Vergata, University of Rome Tor Vergata. Peanut allergy was defined by the following criteria: (a) history of significant peanut-related clinical symptoms, (b) positive skin prick test to peanut allergen (wheal ≥ 3 mm larger than the saline control), and (c) positive in vitro serum peanut IgE (CAP-FEIA) > 0.1 Ku/L. Exclusion criteria were as follows: (a) BMI > 25 Kg/mq, (b) other concomitant food allergies, and (c) autoimmune diseases. A final number of 9 peanut-allergic patients (8 males), age range 4.1–21.7 years, were included in the study. These criteria allowed appropriateness of patient enrollment in the study. Further, blood samples for the lipidome profile of red blood cell membranes were obtained ≥18 months since last allergic reaction to standardize time of testing and reduce the risk of interfering factors. Component-resolved diagnostics (CRDs) for Arah2 and Arah9 were available in 4 patients. As healthy controls, 15 healthy individuals (8 females and 7 males) with BMI < 25 and age 17 ± 4 years old were selected from an anonymous database available in the Lipidomic Laboratory, for which informed consent was gathered at the moment of blood withdrawal, and respect of EU general data protection regulation 2016/679 (GDPR) was ensured by the ISO 17025 certification. All controls did not have any history of allergic reactivity. The benchmark of reference interval values for mature RBC membranes served for observing controls and patients using the data reported for populations [15,16,17].

### 4.2. Isolation of Fatty Acids from RBC Membrane Phospholipids and Gas Chromatographic Analysis

Fatty-acid-based membrane lipidome analyses were performed by the Lipidomic Laboratory of Lipinutragen (Bologna, Italy). Blood samples (0.5 mL) collected in vacutainer tubes with ethylenediaminetetraacetic acid (EDTA) were treated according to the ISO17025 certified procedure (accredited Lab. #1836L) by robotic equipment and processed as described in previous studies [21,22,36,46]. Briefly, the mature cell fraction was isolated based on the higher density of the aged cells with control of diameter controlled by cell counter (Scepter 2.0 with Scepter Software Pro, EMD Millipore, Darmstadt, Germany) [47]. After phospholipid extraction, derivatization to fatty acid methyl esters (FAME) was performed, transforming membrane glycerophospholipids (mainly phosphatidylcholine, phosphatidylethanolamine, phosphatidylserine, phosphatidyl inositol, and plasmalogens) to examine up to 80% of the RBC membrane lipidome [48]. Fatty acids analysis was performed by gas chromatography (GC), and percentages are given as % relative quantitative (% rel. quant.), as previously described [21,22,36,46], comparing with the benchmark of the interval values of each fatty acid and index [15]. 

### 4.3. Statistical Analysis

Statistics were performed using GraphPad Prism 8.0 software (GraphPad Software, Inc., San Diego, CA, USA), applying unpaired *t*-test and Spearman correlation with a 95% confidence interval.

## Figures and Tables

**Figure 1 ijms-24-00120-f001:**
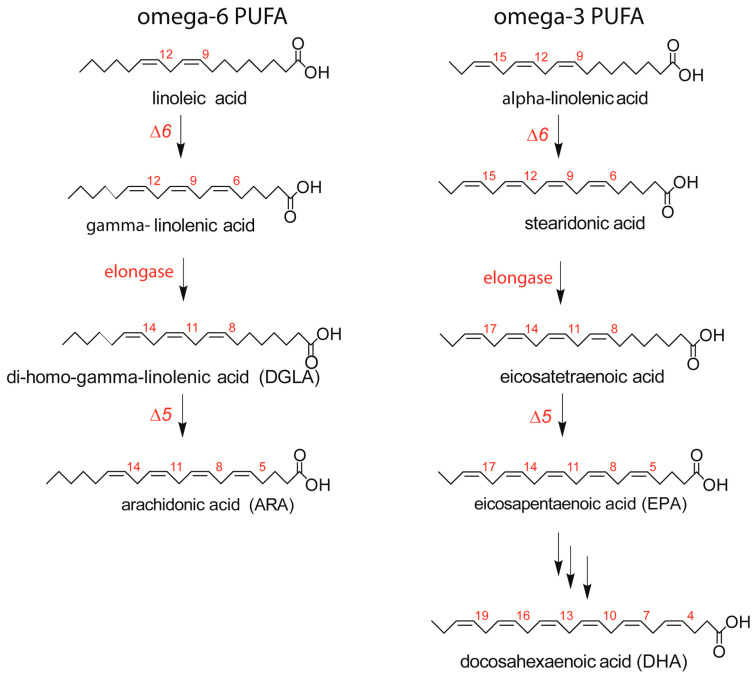
The omega-6 (ω-6) and omega-3 (ω-3) pathways of main long-chain polyunsaturated fatty acid (LCPUFA) biosynthesis, with the interplay of desaturase and elongase enzymes starting from essential (dietary) fatty acid precursors.

**Figure 2 ijms-24-00120-f002:**
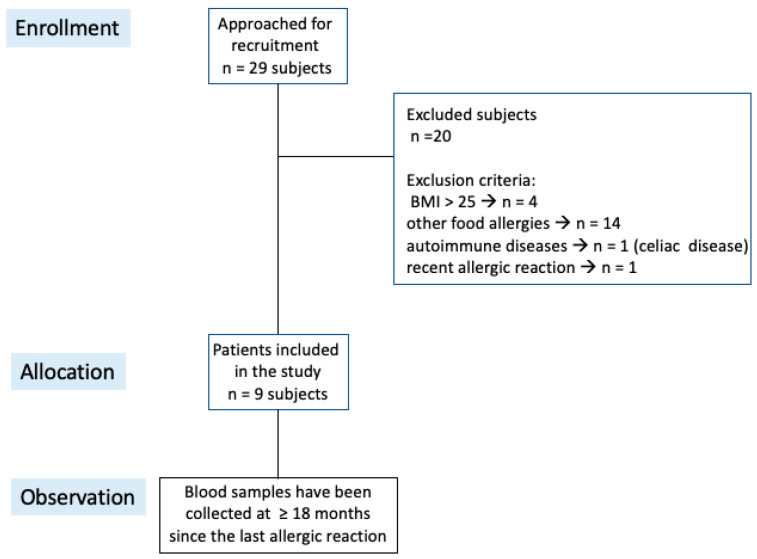
Consort diagram of the patients in this study.

**Figure 3 ijms-24-00120-f003:**
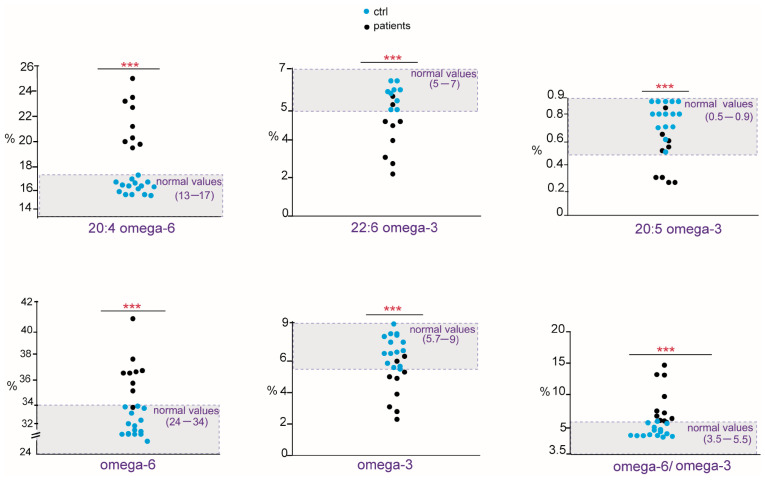
Scattered dot plots of fatty acids, families, and indexes significantly changed in peanut-allergic patients (black dots) compared with healthy controls (blue dots). Data are reported in Table 2. The benchmark of interval values in healthy population is highlighted in light gray in the graphs. *p*-values, *** ≤ 0.0001.

**Figure 4 ijms-24-00120-f004:**
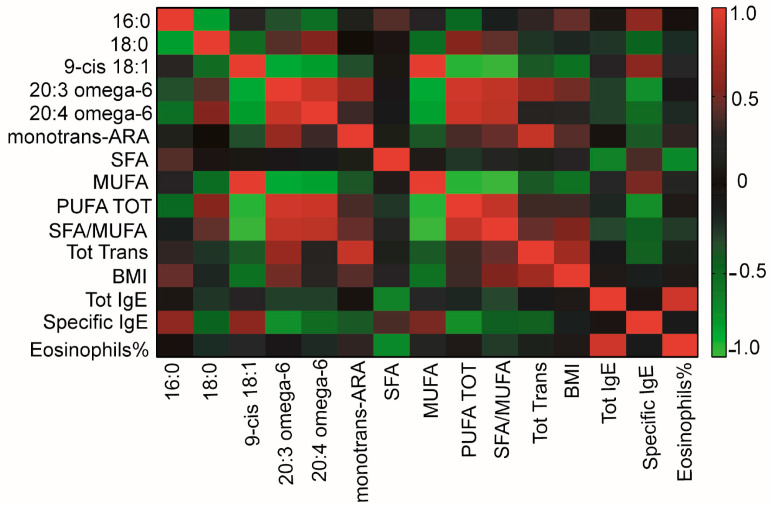
Heatmap showing correlations between patients’ clinical and laboratory data (Table 1) and fatty acid values (Table 2).

**Figure 5 ijms-24-00120-f005:**
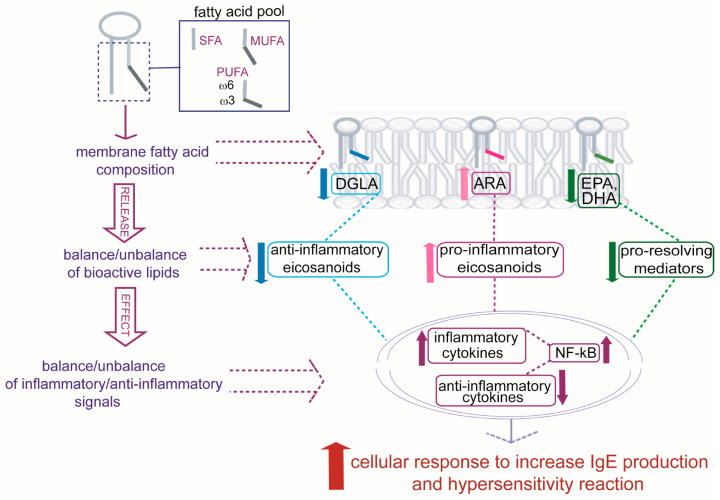
The individual fatty acid pool, derived from diet and metabolism, influences the formation of phospholipids, and an unbalance occurs in allergic patients among dihomo-gamma-linolenic acid (DGLA), arachidonic acid (ARA), eicosapentaenoic acid (EPA), and docosahexaenoic acid (DHA) residues present in cell membranes. The fatty acid composition has impact on formation of bioactive lipids and balance/unbalance of transcription factors (NF-*k*B, among others) for the production of inflammatory/anti-inflammatory cytokines influencing the cellular IgE production and the final hypersensitivity reaction.

**Table 1 ijms-24-00120-t001:** Peanuts allergic patient characteristics.

Sex	Age (Years)	BMI (Centil)	BMI (Kg/mq)	Other Food Allergies	Inhalant Allergies	Reaction on Exposure to Peanuts	Allergic Comorbidity *	Totals IgE (UI/mL)	Peanut-Specific IgE (UI/mL)	Molecular Diagnosis (kU/L)
M	4.6	80	18	-	+	ANAPHYLAXIS	R	965	>100	100 (Arah2)
M	11.7	80	22.7	-	+	ANGIOEDEMA	A-R-AD	965	0.50	NA
M	10.1	75	20.5	-	-	ANGIOEDEMA	R-AD	39.5	32.5	NA
M	21.7	96	28.7	Walnut	+	ANAPHYLAXIS	R-C	152	1	10.3 (Arah9)
M	13.1	40	19.7	-	+	ANAPHYLAXIS	A-R	340	0.54	0.27 (Arah2)
M	17.6	77	24.5	Walnut	+	ANAPHYLAXIS	A-R-C	587	5.29	6.97 (Arah9)
F	12.7	80	23.9	-	+	ANAPHYLAXIS	R-AD	276	>100	NA
M	12.4	10	16.5	Hazelnut	-	ANGIOEDEMA	none	34.5	2.96	NA
M	4.1	75	17.2	Hazelnut	+	ANGIOEDEMA + INTESTINAL symptoms	none	1833	48.8	NA

* A, asthma; R, rhinitis; C, conjunctivitis; AD, atopic dermatitis; NA, not available.

**Table 2 ijms-24-00120-t002:** Fatty acid profile of mature red blood cell membranes of peanut-allergic patients (*n* = 9) in comparison with the reference interval values of healthy population reported in refs. [15,16,17] and with age- and BMI-matched healthy controls (n = 15).

Fatty Acids ^†^	Reference Interval Values	Healthy Controls (n = 15) ^‡^ (% rel. quant.)	Patients (n = 9) ^‡^ (% rel. quant.)
16:0	17–27	22.78 ± 1.02	22.03 ± 1.26
9-*cis* 16:1	0.2–05	0.26 ± 0.11	0.20 ± 0.06
18:0	13–20	18.04 ± 1.36	16.57 ± 1.07 *
9-*trans* 18:1	0.1–0.3	0.11 ± 0.03	0.20 ± 0.03 ***
9-*cis* 18:1	9–18	18.70 ± 1.39	18.83 ± 2.26
11-*cis* 18:1	0.7–1.3	1.03 ± 0.15	1.04 ± 0.18
18:2 ω-6	9–16	13.45 ± 1.14	12.96 ± 1.20
20:3 ω-6 (DGLA)	1.9–2.4	2.06 ± 0.44	1.95 ± 0.35
20:4 ω-6 (ARA)	13–17	16.19 ± 0.64	21.68 ± 1.9 ***
mono-*trans* ARA	0.1–0.4	0.07 ± 0.05	0.14 ± 0.06 **
20:5 ω-3 (EPA)	0.5–0.9	0.78 ± 0.12	0.47 ± 0.21 ***
22:6 ω-3 (DHA)	5–7	6.54 ± 1.03	3.92 ± 1.27 ***
SFA	30–45	40.82 ± 1.34	38.60 ± 0.65 ***
MUFA	13–23	19.99 ± 1.46	20.07 ± 2.35
PUFA ω-6	24–34	31.69 ± 1.65	36.59 ± 1.93 ***
PUFA ω-3	5.7–9	7.32 ± 1.12	4.39 ± 1.43 ***
PUFA TOT	28–39	39.01 ± 1.64	40.98 ± 2.30 *
SFA/MUFA	1.7–2	2.05 ± 0.18	1.94 ± 0.21
ω-6/ω-3	3.5–5.5	4.44 ± 0.84	9.31 ± 3.47 ***
Tot Trans	≤0.4	0.18 ± 0.05	0.35 ± 0.07 ***
PI	138–151	139.81 ± 8.15	129.95 ± 12.04 *

^†^ Fatty acids are reported from the gas chromatographic analysis (GC) after transformation of membrane phospholipids of mature RBCs into fatty acid methyl esters (FAMEs); ^‡^ values are expressed as relative percentages (mean ± SD) of the quantitative values of each fatty acid obtained by calibration curves of the standards and referred to the representative 10 fatty acids cohort. Statistical analyses comparing patients with healthy controls gave the following *p*-values: * ≤0.026; ** ≤0.0022; *** ≤0.0001.

**Table 3 ijms-24-00120-t003:** Correlation (expressed by the Spearman correlation coefficient, r) of the fatty acid values of Table 2 with the peanut-specific IgE shown in Table 1. See also heatmap in Figure 4.

Correlation	r (+/−)	*p*-Value
16:0—specific IgE	0.6	0.086
18:0—specific IgE	−0.469	0.205
9-cis 18:1—specific IgE	0.5	0.1
20:3 ω-6—specific IgE	−0.724	0.033 *
20:4 ω-6—specific IgE	−0.517	0.162
Monotrans-ARA—specific IgE	−0.3	0.391
SFA—specific IgE	0.377	0.30
MUFA—specific IgE	0.519	0.155
Total PUFA—specific IgE	−0.711	0.038 *
Tot trans—specific IgE	−0.444	0.235

* Statistically significant correlations.

## Data Availability

Not applicable.

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
