# Peer review of "Fatty-Acid-Based Membrane Lipidome Profile of Peanut Allergy Patients: An Exploratory Study of a Lifelong Health Condition"

_ijms, 2022, doi:10.3390/ijms24010120_

Round 1

Reviewer 1 Report

In this manuscript Authors performed the lipidome profile of peanut allergy patients. There are many issues with the study design and interpretation with a very low power of study and lack of basic experiemtns substantiating some of the key takeaways of the study. Following are some major concerns.

1. Authors should provide an inclusion and exclusion criteria for the patient including the demography profile and any chronic or acute illness other than allergy. 

2. Authors should provide a Cosort Diagram for easy understanding of the study as it was conducted. 

3. Authors should compare these results with a normal non allergic population. 

4. It is highly desirable to show with the series of experiments the resolution of some of the fatty acids after the crisis. 

5. Authors should also perform some proof of the concept studies in vitro or in vivo on mouse model to confirm their findings

Author Response

In this manuscript Authors performed the lipidome profile of peanut allergy patients. There are many issues with the study design and interpretation with a very low power of study and lack of basic experiemtns substantiating some of the key takeaways of the study.

We are perfectly aware of the small cohort considered in this study as mentioned in the title, abstract and along all manuscript. We appreciate that this referee considered our results giving some major points to ameliorate, meaning that he/she has recognized the novelty of our data regarding polyunsaturated fatty acids in the scenario of the specific disease. We provide point by point answers to his/her observations.

Following are some major concerns.

  1. Authors should provide an inclusion and exclusion criteria for the patient including the demography profile and any chronic or acute illness other than allergy. 

We added in the study subjects (Materials and Methods par 4.1) that the patients are all Italian native patients; other demographic details are reported in Table 1. We also remarked other characteristics of the patients in line with the point 2 of this referee (see below).

  1. Authors should provide a Cosort Diagram for easy understanding of the study as it was conducted. 

Indeed, this suggestion of the referee was very important, since we considered more subjects at the beginning but then we chose only 9 of the initial group to focus on patients with normal BMI and far from the allergic episode, so that the erythrocyte membrane profile could not be attributed to other causes such as overweight or acute responses. A new Figure 2 is introduced as Consort diagram.

  1. Authors should compare these results with a normal non allergic population. 

The reference interval values in Table 2 are reported from literature values and an Italian population cohort (refs 15,17) which are healthy and exclude allergic subjects. The benchmark to compare the values of patients’ cohorts has been specified in the text. 

  1. It is highly desirable to show with the series of experiments the resolution of some of the fatty acids after the crisis. 

In this study we excluded subjects that were <18 months distant from the last crisis (see Figure 2). Another interesting point will be the evaluation of the FA values before-after the crisis, and this is an interesting suggestion to deepen the scenario of allergic diseases.

  1. Authors should also perform some proof of the concept studies in vitro or in vivo on mouse model to confirm their findings

Our study suggests the precise involvement of DGLA and DHA as polyunsaturated fatty acids in food allergy. We think that this result can motivate further studies with cell and animal models connecting the allergic molecular mechanism with specific fatty acids and the bioactive lipids formed in their metabolism. In the Discussion (line 252-254) we mentioned ref 34 with mice model.

Reviewer 2 Report

In their manuscript entitled “Fatty acid-based membrane lipidome profile of peanut allergy patients: an exploratory study of a life-long health condition” Del Duca et al., provide information regarding the lipidome profile of red blood cell membranes in peanut allergy patients. The study is interesting and well-written, and despite the low number of samples and assays, it provides novel and important data for the scientific community. However, this Reviewer has some comments as follow:

Introduction section:
It would be beneficial for the article if the Introduction section was split into more paragraphs (e.g., the first paragraph could end at line 46 and concern the general information regarding peanut allergy, the second could focus on lipids and include lines 46-54 etc.). This will make the text easier to follow and help the reader digest the information.

Methods section:
The authors state that they compare the values of FAME analysis of the allergy samples with control samples that have been previously reported. Since these samples were ran at a different time compared to the controls, are they indeed comparable? Was any internal control used? This question is based on the fact that in most techniques/assays it is important to simultaneously measure the groups under comparison. This Reviewer a priori apologizes if he/she misunderstood the way these data were evaluated.

In this Reviewer's opinion the study would be more complete if more parameters were assessed, including lipid peroxidation. While the peroxidation index implies the susceptibility of membrane lipid to oxidation it would be interesting to really detect the level of oxidation in allergic subjects vs control (e.g., via MDA detection). In this Reviewer's opinion this should be mentioned as a limitation of this study.

Results section:
Figure 3: Please indicate the color of positive and negative correlations in the Figure to facilitate the reader.

Discussion section:
While the discussion is extensive and adequately supported by relative references, there is not a lot of information focused on the impact, mechanism, role or reason of the specific lipid alterations in peanut allergic subjects. The discussion is rather general than focused on the specific topic. This Reviewer believes it would be important to differently construct this part of the study to make it clearer what the main result of this study is. Especially regarding how the RBC lipidomic alterations the authors found impact the immunological responses.  

Author Response

In their manuscript entitled “Fatty acid-based membrane lipidome profile of peanut allergy patients: an exploratory study of a life-long health condition” Del Duca et al., provide information regarding the lipidome profile of red blood cell membranes in peanut allergy patients. The study is interesting and well-written, and despite the low number of samples and assays, it provides novel and important data for the scientific community.

We thank this referee for his/her appreciation of our work.

However, this Reviewer has some comments as follow:

Introduction section:
It would be beneficial for the article if the Introduction section was split into more paragraphs (e.g., the first paragraph could end at line 46 and concern the general information regarding peanut allergy, the second could focus on lipids and include lines 46-54 etc.). This will make the text easier to follow and help the reader digest the information.

We reorganized the introduction as requested by this referee. The Introduction has now two sections:

1.1 Fatty acids as structural and functional constituents of cell membranes and relationship with diet

1.2 Fatty acid-based membrane lipidome analysis

Methods section:
The authors state that they compare the values of FAME analysis of the allergy samples with control samples that have been previously reported. Since these samples were ran at a different time compared to the controls, are they indeed comparable?

The analyses of mature erythrocytes in population studies and in the present study are performed using the same protocol, active in Lipidomic Laboratory at the Lipinutragen company, following a procedure that has received the ISO17025 accreditation (laboratory method), yearly audited by the Italian Body of Accreditation (Accredia). We specified in the Results section (lines 115-125) that the data are reported after comparison with reference compounds and calibration to give quantitative data. The quantity of each fatty acid is reported as percentage relative to the total of the fatty acid quantities detected in the sample (% rel. quant.) and, together with the quality assurance for the equipment functioning and procedures, this renders each sample comparable to others and to different measurements done in different times in our Lipidomic Laboratory.

Was any internal control used? This question is based on the fact that in most techniques/assays it is important to simultaneously measure the groups under comparison. This Reviewer a priori apologizes if he/she misunderstood the way these data were evaluated.

This is a motivated concern of the referee since the analyses must be comparable and this can be done through internal standards, if there are problems of identification, and by calibration procedures to calculate fatty acid quantities. We follow calibrated procedures and we detailed this in the Results and in the Experimental sections.  For biological samples, two conditions must be met: clear separation of geometrical and positional isomers, so that the peaks are not superimposed, and the presence of adequate standard mixtures for recognition and calibration. In lines 285-300 of the Discussion this aspect has been pointed out.

In this Reviewer's opinion the study would be more complete if more parameters were assessed, including lipid peroxidation. While the peroxidation index implies the susceptibility of membrane lipid to oxidation it would be interesting to really detect the level of oxidation in allergic subjects vs control (e.g., via MDA detection). In this Reviewer's opinion this should be mentioned as a limitation of this study.

We added this point in the Discussion section (lines 280-284) mentioning the limitation of not having a direct measure of lipid peroxidation. Peroxidation index is an accepted measure of the peroxidizability of the biological environment. It was useful to point it out, since we found a recent review where direct measurements were discussed (Murphy, M.P., Bayir, H., Belousov, V. et al. Guidelines for measuring reactive oxygen species and oxidative damage in cells and in vivo. Nat Metab 4, 651–662 (2022)). 

Results section:
Figure 3: Please indicate the color of positive and negative correlations in the Figure to facilitate the reader.

WE added the legend in the Figure.

Discussion section:
While the discussion is extensive and adequately supported by relative references, there is not a lot of information focused on the impact, mechanism, role or reason of the specific lipid alterations in peanut allergic subjects. The discussion is rather general than focused on the specific topic. This Reviewer believes it would be important to differently construct this part of the study to make it clearer what the main result of this study is. Especially regarding how the RBC lipidomic alterations the authors found impact the immunological responses.  

Thank you very much for this important suggestion. In the Introduction we created Section 1.2 on the importance of RBC membrane for the representation of fatty acids in tissues. We added a final scenario with references and a  new Figure (Figure 5) to express better the impact of the specific fatty alterations in the allergic reactivity.

Round 2

Reviewer 1 Report

Although authors have sincerely attempted to resolve all the issues one major issue still remains in the manuscript i.e. the authors have not compared their population with the healthy and matched normal control population. This is very important concern to address since you would want to show a signficant deviation in patient population with the parameters which you are measuring in your established assay in comparision to healthy and matched controls. These results cannot be compared with the published healthy control literature. I strongly recommend authors to compare the results of the study with the matched healthy control population.

Author Response

We thank the referee for his/her suggestion. We added the results of 15 healthy controls, selected for age and BMI to match the patients’ cohort.

MODIFIED TABLE 2: Now in Table 2 there are two columns for patients and healthy controls and we performed statistical analysis, indicating the significance by p values in the Table. We did not cancel the benchmark of interval values available in population studies, to offer a larger overview of published data for healthy population, especially for evaluating the importance of measuring the fatty acid values that we reported in much smaller healthy and unhealthy cohorts.

MODIFIED Figure 3: we added the values of the healthy cohort (n=15)

DISCUSSION: We modified the Discussion with the comments related to the 15 healthy controls, in particular evidencing new aspects of comparison with our allergic cohort, such as the significant diminution of EPA and increase of trans fatty acids (that we had also evidenced in a former work done in 2005 and now mentioned as ref. 40). We underlined that the study remains preliminary due to the small number of subjects considered. We introduced in the discussion 4 new references.

We added an EXCEL file with the raw data of patients and healthy cohort for your examination (not publishable data).

Reviewer 2 Report

The authors have properly addressed all my concerns. I think this manuscript will be of great value for the scientific community.

Author Response

Thank you very much for your appreciation.

Round 3

Reviewer 1 Report

The authors have addressed all the concerns and I have no further comments.